# Semantic-Centric Alignment for Zero-shot Panoptic and Semantic Segmentation

## Abstract

Zero-shot segmentation has achieved great success by generating features from semantic embeddings to adapt the model to unseen classes. These semantic-generated features are typically aligned with the visual distribution of seen classes to improve generalization on extracted image features. However, this vision-centric alignment may easily overfit seen classes due to the lack of visual data for unseen classes. To address this issue, we propose a semantic-centric alignment method that aligns the generated features with the well-structured semantic distribution across all classes. First, we align the vision backbone features with CLIP tokens through Vision-to-CLIP alignment. This approach leverages CLIP's visual-language matching capabilities to produce semantic-aligned backbone features. Then, we generate synthetic features from semantic embeddings for unseen classes, supervised by semantic-aligned visual features and CLIP semantic tokens for improving visual diversity while maintaining semantic consistency. Finally, we finetune the class projector through the semantic-aligned joint features to further adapt the model for unseen classes. Our semantic-centric alignment effectively enhances the model's zero-shot generalization by constructing a unified and well-structured semantic-aligned feature space. Our method achieves SOTA performance in both zero-shot panoptic and semantic segmentation, and can directly segment unseen classes without fine-tuning.

## 1 Introduction

Semantic segmentation, a critical task in computer vision, has seen remarkable success driven by advances in deep learning (He et al., 2016; Vaswani et al., 2017). However, obtaining high performance demands massive data with detailed pixel-level annotations, which are labor-intensive and expensive to generate. This challenge has driven segmentation research into zero-shot segmentation (Chen et al., 2023; Gu et al., 2020; Bucher et al., 2019).

Zero-shot segmentation aims to generalize the model trained on seen data to segment unseen (novel) classes during inference, relying solely on textual descriptions. The primary challenge is the lack of visual data for unseen classes, which leads to overfitting to the visual distribution of seen classes. To alleviate this problem, existing methods typically adopt a vision-centric approach, training a generator to produce synthetic visual features for adapting the mododel to unseen classes (Bucher et al., 2019; Cheng et al., 2021b; Gu et al., 2020), as shown in Fig. 1. Formally, the generator maps semantic embeddings to the visual space and aligns the generated visual features with real visual features extracted from the visual backbone.

However, due to the absence of visual data for unseen classes, even though the visual backbone captures sufficient visual attributes (Ding et al., 2022a; Han et al., 2023a), it still struggles to establish a well-structured visual feature space for properly positioning unseen classes. Consequently, the vision-aligned feature generator tends to bias toward seen classes when generating synthetic features, hindering generalization to unseen classes. Although some methods attempt to guide visual feature generation through semantic relationships (He et al., 2023b; Han et al., 2023b), the established visual feature space still remains imperfect due to the modality gap. Thus, it is often suboptimal to generalize new classes in the under-constructed visual space of vision-centric approaches.

In this paper, we propose a novel semantic-centric method to align the backbone and generator features with the well-structured semantic distribution, ensuring all classes are properly positioned due

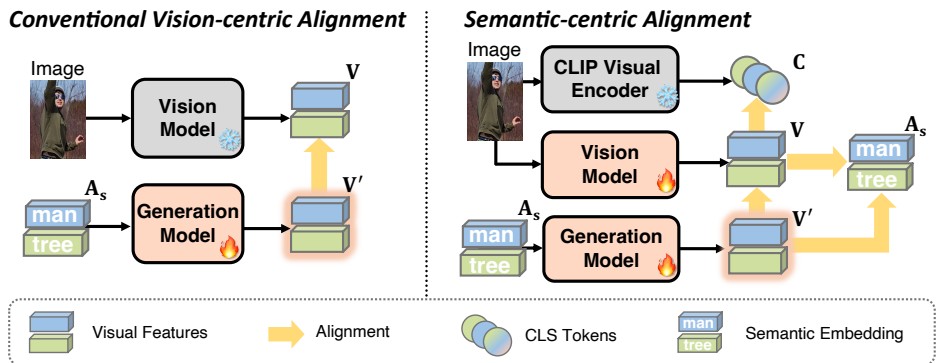

Figure 1: Comparisons between conventional vision-centric alignment and our semantic-centric alignment. *Left*: Vision-centric alignment, which aligns the generator with vision models trained solely on seen visual data. *Right*: Our semantic-centric alignment, which aligns the backbone and generator features with well-structured semantic distribution learned from CLIP.

to the highly optimized CLIP semantic embeddings. We begin with improving the visual backbone's semantic alignment by aligning its features with CLIP-extracted tokens through Vision-to-CLIP (V2C) alignment. Next, we generate synthetic features for unseen classes, supervised by semantic-aligned visual backbone features and CLIP semantic tokens, to improve visual diversity while maintaining semantic consistency. This Generation-to-Semantic (G2S) alignment ensures that the generated synthetic features are appropriately positioned in the well-optimized semantic feature space for improved generalization to unseen classes. Finally, we adapt the feature projector to collaborate the real and synthetic features for recognizing both seen and unseen classes.

Our approach fundamentally differs from conventional vision-centric zero-shot segmentation methods. For example, ZS3 (Bucher et al., 2019) and CaGNet (Gu et al., 2020) focus primarily on learning visual distributions through the visual feature generator, while Joint (Baek et al., 2021) projects visual and semantic features into a joint space to enhance visual feature optimization, and PADing (He et al., 2023b) uses semantic relationships to guide visual feature generation. In contrast, our approach transfers knowledge from the visual to the semantic feature space through semantic-centric alignment, leveraging the highly optimized semantic distribution to facilitate high-quality feature generation and zero-shot generalization for unseen classes.

Thus, our method is unique in its idea and design on exploring the well-structured semantic distribution for feature alignment, thereby addressing the overfitting problem caused by limited visual data. Furthermore, our method is simple and can be flexibly integrated into existing powerful segmentation models, such as Maskformer (Cheng et al., 2021a) and Mask2former (Cheng et al., 2022), and achieves state-of-the-art performance on multiple zero-shot panoptic and semantic segmentation benchmarks. Extensive experiments and analyses demonstrate the effectiveness of our semantic-centric method, proving the substantial generalization capability of the well-structured semantic-aligned feature space. The semantic-centric property enables the model to directly segment unseen classes in the inductive setting without fine-tuning, as the semantic-aligned backbone features effectively match the semantic embeddings for new concepts. In summary, our key contributions are:

– We propose a novel semantic-centric alignment method to align features with the well-structured semantic distribution for zero-shot segmentation.

– We successfully achieve the semantic-centric alignment by collaboratively aligning the backbone features and generated features, and adapting feature projector for zero-shot generalization.

– Our method achieves state-of-the-art performance on multiple zero-shot panoptic and semantic segmentation benchmarks, and can be flexibly integrated into powerful segmentation models.

## 2 RELATED WORKS

**Close-set Image Segmentation.** Image segmentation, a fundamental task in computer vision, involves categorizing each component of an image. It can be categorized into three types: semantic

segmentation (Long et al., 2015; Chen et al., 2018), where pixels are classified into categories; instance segmentation (He et al., 2017; Tian et al., 2020), which distinguishes individual objects or instances; and panoptic segmentation (Li et al., 2021; Kirillov et al., 2019), a hybrid approach that classifies both background pixels and foreground instances. Panoptic segmentation has garnered significant interest from the research community due to its comprehensive approach that addresses the challenges of both semantic and instance segmentation simultaneously.

Before the wide application of transformer (Vaswani et al., 2017; Dosovitskiy et al., 2020) in computer vision, many works (Li et al., 2021; Xiong et al., 2019) treat panoptic segmentation as the combination of instance segmentation and semantic segmentation. Since the proposal of DETR (Carion et al., 2020), panoptic segmentation entered a new era. In a specific, first, the model is replaced by self-attention (Vaswani et al., 2017; Dosovitskiy et al., 2020) rather than CNN-based. Moreover, the things and stuff are modeled together based on the Hungarian algorithm (Zhang et al., 2021; Li et al., 2022b; Cheng et al., 2021a; 2022; Carion et al., 2020; Li et al., 2023a) rather than separately processed. In this paper, the proposed method can handle both semantic and panoptic segmentation in a more challenging zero-shot scenario.

**Zero-shot classification.** Zero-shot classification aims to recognize classes that are not present in the training dataset. This task, characterized by its challenging nature and significant potential for real-world applications, has attracted increasing attention from researchers. Common approaches utilize attribute-based datasets, such as AWA (Lampert et al., 2009) and CUB (Wah et al., 2011), where each category is defined by distinct attributes, *e.g.*, seals are described as *furry*, *big*, and capable of *swimming* (Han et al., 2020; Su et al., 2022; Wu et al., 2020; Han et al., 2021; Chen et al., 2021; 2020; Zhu et al., 2019). Typically, these methods begin by training a generator to synthesize visual features for unseen categories. Subsequently, they generate several pseudo-unseen visual features based on attributes of unseen classes. Finally, a trainable classifier (Han et al., 2020; Su et al., 2022; Wu et al., 2020; Han et al., 2021; Chen et al., 2021) or a k-NN classifier (Han et al., 2020; Chen et al., 2020; Zhu et al., 2019) is employed on the test dataset. Although effective, zero-shot classification neglects valuable visual cues critical for segmentation tasks. Besides, the zero-shot segmentation only contains the name of a category without any attribute description. While not directly applicable to segmentation, the insights gained from zero-shot classification methods are inspiring.

**Zero-shot Segmentation.** Most current zero-shot segmentation approaches can be classified into two categories: projection-based (Xian et al., 2019; Ding et al., 2022a; Zhou et al., 2023; Xu et al., 2022; Chen et al., 2023; Baek et al., 2021; Zhang & Ding, 2021) and generation-based methods (He et al., 2023b; Cheng et al., 2021b; Baek et al., 2021; Pastore et al., 2021; Gu et al., 2020; He et al., 2023a). Projection-based methods align visual and semantic features within the same space and calculate the similarity between them, assigning the category with the highest similarity score. Conversely, generation methods first train a generator to synthesize unseen visual features. A trainable classifier is then developed to work with both real-seen and generated unseen visual features, effectively transforming zero-shot segmentation into traditional segmentation methods. Additionally, recent efforts have explored open-vocabulary segmentation (Xu et al., 2023b; Ding et al., 2022b; Xu et al., 2023a; Qin et al., 2023; Han et al., 2023a; Xu et al., 2022; Gu et al., 2022; Li et al., 2023b; Ma et al., 2022; Han et al., 2023b; Zhang et al., 2024), which, like zero-shot segmentation, aims to recognize unseen categories but involves training on a complete dataset, unlike the more limited data used in zero-shot approaches.

In this paper, we introduce a novel cyclic alignment method for zero-shot segmentation, adhering to the generation approach. Unlike traditional generation methods, our strategy, *i.e.*, S2V alignment, enhances the visual diversity of the generated unseen visual features, thereby capturing a more accurate visual distribution. Furthermore, while most methods do not reproject generated features back to the semantic space, potentially compromising semantic generalization and restricting category flexibility, our V2S alignment maps the generated visual features back to the semantic space and aligns with the semantic features, preserving generalization capabilities.

## 3 METHOD

**Method Overview** Our model illustrated in Fig. 2, consists of a backbone based on transformer decoder models, such as Mask2Former (Cheng et al., 2022), whose outputs are a group of vectors without spatial dimensions. It also includes a class projector, composed of MLPs, which replaces

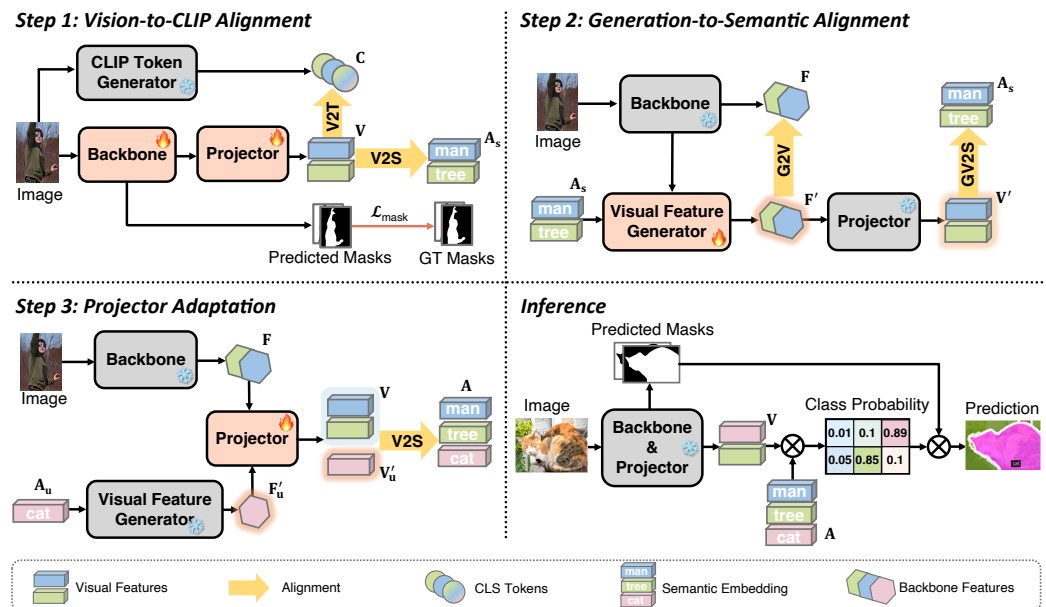

Figure 2: The overall training procedure of our method. First, train the visual backbone and the projector by the proposed Vision-to-Token (V2T) and Vision-to-Semantic (V2S) alignment to align with both CLIP visual and textual encoder, obtaining a semantic-aligned visual backbone. Next, we freeze the backbone and the projector, and train a generator to synthesize backbone features and visual features and align with their real counterparts, resulting in a visually diverse and semantically consistent feature generator. Finally, we freeze the backbone and the generator, and sample real seen and synthetic unseen, respectively, which are used to adapt the class projector for unseen classes. During inference, due to the great generalization, our method can directly segment new classes by adding new semantic embeddings rather than retraining the classifier like conventional methods.

the learnable classifier to align the dimensions between the visual embeddings and the semantic embeddings $\mathbf{A}$ (including seen categories $\mathbf{A}_s$ and unseen categories $\mathbf{A}_u$). Additionally, a feature generator is used to produce synthetic visual features for unseen classes from the corresponding semantic embeddings. The training of our method involves three stages. *First*, we apply vision-to-CLIP alignment to train the visual backbone and class projector by aligning the visual features with both CLIP semantic embeddings and visual CLS tokens. This alignment transfers vision-language matching capabilities from CLIP to the backbone, leading to a semantic-aligned visual backbone. *Second*, we freeze the backbone and class projector and introduce Generation-to-Semantic alignment to train the generator. In addition to aligning the generated visual features with the frozen semantic-aligned backbone, we encourage the generator to produce more diverse synthetic visual features while maintaining semantic consistency with the input embeddings. *Finally*, we freeze both the backbone and generator to produce real seen and pseudo unseen features, adapting the class projector for unseen classes.

## 3.1 VISION-TO-CLIP ALIGNMENT FOR BACKBONE ADAPATION

The core idea is to align the visual features with both the CLS tokens, which are well-aligned with generalizable semantics from CLIP's visual encoder, and the semantic embeddings, which are aligned with real visual distributions from CLIP's textual encoder, as shown in the top left of Fig. 2. Vision-to-CLIP (V2C) alignment consists of two components: Vision-to-Semantic (V2S) alignment and Vision-to-Token (V2T) alignment. The goal of V2S alignment is to match visual features with their corresponding semantic embeddings to ensure that the backbone's visual features are aligned with accurate semantics. Given an input $\mathbf{X}$ and its label set $\mathbf{Y} = \{\mathbf{y}_i\}_i^O$, where each $\mathbf{y}_i$ represents non-overlapping segments and $O$ is the number of unique objects, we first pass $\mathbf{X}$ through the backbone and class projector to obtain the output $\mathbf{Z} = \{\mathbf{z}_i\}$. Each $\mathbf{z}_i = \{\mathbf{v}_i, \mathbf{m}_i\}$ consists of visual features $\mathbf{v}_i \in \mathbf{V}^{Q \times C}$ and mask predictions $\mathbf{m}_i \in \mathbf{M}^{Q \times H \times W}$. Here, $\mathbf{V}$ and $\mathbf{M}$ represent the visual features and mask predictions, respectively, while $Q$ is the total number of queries. We then

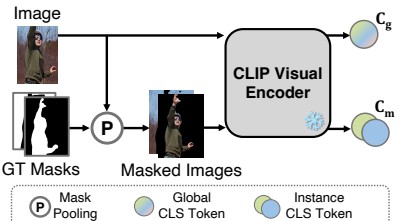 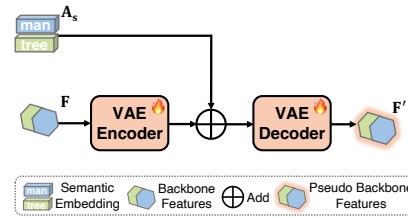

(a) Global and instance CLS token generation.  (b) Synthtic visual feature generation.

Figure 3: The procedures of (a) synthetic visual feature generation and (b) CLS token generation.

compute the similarity scores between $\mathbf{V}$ and the seen semantic embeddings $\mathbf{A}_s$: $\mathbf{P} = \mathbf{S}^\top \mathbf{A}_s$. Since there are no spatial dimensions, we use Hungarian matching (Carion et al., 2020; He et al., 2023b) to find the target assignment $\hat{\sigma}$ that minimizes the matching loss $\mathcal{L}_{match}$ between $\mathbf{Z}$ and $\mathbf{Y}$,

$$\mathcal{L}_{match}(y_{\sigma(\mathbf{z})}, \mathbf{z}) = \mathcal{L}_{focal}(p_{\sigma(\mathbf{z})}, p) + \mathcal{L}_{mask}(m_{\sigma(\mathbf{z})}, m), \tag{1}$$

where $\mathcal{L}_{focal}$ represents the focal loss (Lin et al., 2017), and $\mathcal{L}_{mask}$ is the same as (Cheng et al., 2022). Finally, using the target assignment $\hat{\sigma}$, we apply $\mathcal{L}_{match}$ to align the visual features with the semantic embeddings and capture visual attributes for generating class-agnostic masks.

While V2S ensures that visual features are aligned with semantic embeddings, this process alone may not fully exploit CLIP's generalization capabilities. Therefore, we introduce Vision-to-Token (V2T) alignment, which consists of target attention to better align with the global CLS tokens and instance alignment to capture semantics that may be overlooked by the global CLS tokens. For V2T alignment, we first generate global and instance-level CLS tokens, as shown in Fig. 3a. Ground truth masks are used to exclude regions unrelated to the target objects. Both the masked and original images are passed through the frozen CLIP visual encoder to obtain the instance CLS tokens $\mathbf{C}_m \in \mathbb{R}^{O \times C}$ and the global CLS token $\mathbf{C}_g \in \mathbb{R}^{1 \times C}$. To leverage the global tokens, we propose target attention, which calculates the similarity $\mathbf{W} \in [0, 1]$ between $\mathbf{C}_g$ and the visual features $\mathbf{V}$, emphasizing the similarities corresponding to $\mathbf{V}_s$, which are assigned to objects under $\hat{\sigma}$.

$$\mathbf{W} = \text{softmax}\Big( \frac{\mathbf{C}_g^\top \cdot \mathbf{V}}{||\mathbf{V}||_2 \cdot ||\mathbf{C}||_2} \cdot (\gamma \cdot \mathbb{1}(\mathbf{v} \in \mathbf{V}_s) + 1) \Big), \mathbf{v} \in \mathbf{V}, \tag{2}$$

where softmax is applied along the second dimension of $\mathbf{W}$, and $\gamma$ is a hyperparameter used to emphasize the importance of features assigned to targets. Based on the target attention, we aggregate $\mathbf{V}$ to obtain $\mathbf{V}_a$, where $\mathbf{V}_a = \mathbf{W}^\top \cdot \mathbf{V}$. Inspired by CLIP-ZSS (Chen et al., 2023), we then use CLS token banks to store $\mathbf{C}_g$ from different images as negative pairs. We concatenate $\mathbf{C}_g$ with the tokens in the CLS token bank to form $\mathbf{C}_b$. Finally, we align the aggregated visual features $\mathbf{V}_a$ with the global CLS tokens $\mathbf{C}_b$,

$$\mathcal{L}_g(\mathbf{V}_a, \mathbf{C}_b) = \frac{\exp(\mathbf{V}_a^\top \cdot \mathbf{c}_i / \tau)}{\sum_{j \neq i}^{T} \exp(\mathbf{V}_a^\top \cdot \mathbf{c}_j) / \tau) + \exp(\mathbf{V}_a^\top \cdot \mathbf{c}_i / \tau)}, \tag{3}$$

where $\mathbf{c} \in \mathbf{C}_b$. Relying solely on global tokens may limit the model's ability to distinguish between objects and could result in overlooking less prominent classes in an image. To address this issue, we propose instance alignment to supervise $\mathbf{V}_s$ using $\mathbf{C}_m$.

$$\mathcal{L}_l(\mathbf{V}_s, \mathbf{C}_m) = \sum_i^O \frac{\exp(\mathbf{v}_i^\top \cdot \mathbf{c}_i / \tau)}{\sum_{j \neq i}^{O} \exp(\mathbf{v}_i^\top \cdot \mathbf{c}_j) / \tau) + \exp(\mathbf{v}_i^\top \cdot \mathbf{c}_i / \tau)}, \tag{4}$$

where $\mathbf{v} \in \mathbf{V}_s$ and $\mathbf{c} \in \mathbf{C}_m$. To recap, Vision-to-CLIP alignment consists of:

$$\mathcal{L}_{v2c} = \mathcal{L}_{match}(y_{\hat{\sigma}(\mathbf{Z})}, \mathbf{Z}) + \mathcal{L}_g + \mathcal{L}_l, \tag{5}$$

V2C alignment transfers CLIP's vision-semantic matching capabilities to our model, ensuring that the visual features are aligned with the well-structured semantics of CLIP embeddings. This provides the generator with a semantically aligned visual distribution and enhances generalization to unseen classes.

## 3.2 Generation-to-Semantic alignment for generator adaptation

Another bottleneck in zero-shot segmentation is the ineffectiveness of the generator in utilizing semantic information. Therefore, we propose Generation-to-Semantic alignment to encourage the generator to effectively utilize the semantic information. The core idea of the generation-to-semantic, as shown in the top right of Fig. 2, is to map semantic features to the visual space and pass the generated visual features through the frozen class projector, which was trained during Vision-to-CLIP alignment. In the semantic-to-vision mapping phase, we align the synthetic visual features with real ones from the frozen, semantic-aligned backbone. In the projection phase, we directly align the projected features with the corresponding semantic embeddings.

Generation-to-Semantic alignment consists of two key components: Generator-to-Vision (G2V) alignment and Generated-Vision-to-Semantic (GV2S) alignment. Specifically, we first pass $\mathbf{X}$ through the frozen backbone to obtain the real semantic-aligned visual features $\mathbf{F}$. Then, we apply $\hat{\sigma}$ to split $\mathbf{F}$ into $\mathbf{F}s \in \mathbb{R}^{O \times C}$, representing backbone features assigned to targets, and $\mathbf{F}\varnothing \in \mathbb{R}^{(Q-O) \times C}$, representing features without assigned targets. Next, we generate the pseudo visual features $\mathbf{F}'$, as shown in Fig. 3b. Formally, $\mathbf{F}_s$ is fed into the VAE encoder (Kingma & Welling, 2013) to obtain the reparameterized output. Then, the corresponding semantic features $\mathbf{A}'_s$ are added to the reparameterized output as semantic conditions. The result is fed into the VAE decoder to generate the pseudo visual features $\mathbf{F}'_s$.

After obtaining $\mathbf{F}'_s$, we apply G2V alignment to align $\mathbf{F}'_s$ with the real visual features $\mathbf{V}_s$. We follow conventional methods (He et al., 2023b; Bucher et al., 2019; Cheng et al., 2021) to align $\mathbf{F}'_s$ with $\mathbf{F}_s$ using the GMMN loss (Li et al., 2015).

$$\mathcal{L}_{mmd} = \sum_{\mathbf{v}'_s \in \mathbf{V}'_s} k(\mathbf{v}'_s, \mathbf{V}_s) + \sum_{\mathbf{v}_s \in \mathbf{V}_s} k(\mathbf{v}_s, \mathbf{V}'_s) - 2 \sum_{\mathbf{v}'_s \in \mathbf{V}'_s} \sum_{\mathbf{v}_s \in \mathbf{V}_s} k(\mathbf{v}'_s, \mathbf{v}_s), \qquad (6)$$

where $k(a,b) = \exp\left(\frac{1}{2\lambda^2}||a-b||^2\right)$ is a kernel function with a bandwidth $\lambda$. Due to the lack of unseen visual data, the backbone captures only a limited portion of the real visual distribution, resulting in less diversity compared to the complete visual distributions. To address this issue, we apply contrastive alignment to supervise $\mathbf{F}'_s$ with $\mathbf{F}$, encouraging the generation of more diverse visual features.

$$\mathcal{L}_{div} = \Sigma_i^O \frac{\exp(\mathbf{f}_i^{'\top} \cdot \mathbf{f}_i / \tau)}{\Sigma_j^O \exp(\mathbf{f}_i^{'\top} \cdot \mathbf{f}'_j)/\tau) + \Sigma_k^{Q-O} \exp(\mathbf{f}_i^{'\top} \cdot \mathbf{f}_k^{\varnothing}/\tau)}, \qquad (7)$$

where $\mathbf{f} \in \mathbf{F}, \mathbf{f}' \in \mathbf{F}', \mathbf{f}^{\varnothing} \in \mathbf{F}^{\varnothing}$ and $\tau$ is the hyperparameter to control the scale of loss. By pushing away from different $\mathbf{F}_s$ the generator knows the difference between objects and pushing away from $\mathbf{V}^{\varnothing}$ forces the generator to produce the features that are more similar to the real visual distributions. To recap, the total loss function of G2V alignment is: $\mathcal{L}_{g2v} = \mathcal{L}_{mmd} + \mathcal{L}_{div}$, G2V alignment helps the generator align with the semantic-aligned backbones with more visual diversity. Although the G2V alignment aligns pseudo-visual features with real semantic-aligned visual features, the generated visual features may not strictly adhere to the original semantic embeddings. Therefore, we propose the GV2S alignment. Specifically, after aligning $\mathbf{F}'$ with $\mathbf{F}$ through G2V alignment, we map $\mathbf{F}'$ through the frozen class projector to obtain the $\mathbf{F}'_s$. Then, we align $\mathbf{F}'_s$ with the all the seen semantic features $\mathbf{A}_s$ through GV2S alignment, $\mathcal{L}_{gv2s} = \mathcal{L}_{focal}(\mathbf{F}'_s, \mathbf{A}_s)$, where $\mathcal{L}_{focal}$ indicates the same loss function in Vision-to-CLIP alignment, which ensures the semantic consistency between the generator and the backbone while ensuring that the semantics can be integrated into the generated visual embeddings. In summary the total loss function for G2S alignment is:

$$\mathcal{L}_{g2s} = \mathcal{L}_{mmd} + \lambda_s \cdot \mathcal{L}_{g2v} + \lambda_v \cdot \mathcal{L}_{gv2s}, \qquad (8)$$

where $\lambda_s$ and $\lambda_v$ are two hyperparameters to control the scale of cyclic alignment. In generation-to-semantic alignment, G2V and GV2S alignments allow the generator to produce visual features that are not only diverse but also semantically consistent, leading to more realistic generalizations for unseen classes.

## 3.3 Class Projector Adaptation

Different from most of the generation methods (He et al., 2023b; Cheng et al., 2021b; Gu et al., 2020), we finetune the class projector rather than a trainable classifier, which leads to weak generalization capability. Specifically, we first fix the backbone and the generator. Then, we randomly

sample $U$ noise features $\mathbf{U} \in \mathcal{N}^{U*D}$ from a standard Gaussian distribution to feed into the VAE encoder. We also randomly sample $U$ unseen semantic features from $\mathbf{A}_u$, which are used as conditions for the frozen VAE decoder to generate pseudo unseen visual features. Finally, these features are fed into the trainable class projector to obtain the pseudo unseen semantic feature $\mathbf{F}'_u$. The class projector is supervised by:

$$\mathcal{L}_f = \mathcal{L}_{focal}(\mathbf{F}, \mathbf{A}_s) + \mathcal{L}_{focal}(\mathbf{F}'_u, \mathbf{A}). \tag{9}$$

### 3.4 Training Objectives and Inference

Our method need three steps of training. First, we train a backbone and a class projector based on the supervision of seen classes with Vision-to-CLIP alignment (Eq. 5). Second, we freeze the backbone and the class projector and train a generator with Generation-to-Semantic alignment (Eq. 8). Finally, we use the real-seen visual features from the backbone and the pseudo visual features from the generator to finetune the class projector with Eq. 9.

During inference, we input the image into the backbone to extract visual features and generate predicted masks. The visual features are then passed through the fine-tuned class projector and the probability of each visual feature is calculated by taking the inner product between the visual features and the semantic features. Finally, we calculate the final result by applying the inner product between the probabilities and the predicted masks.

## 4 Experiments

### 4.1 Experiments Setup

**Dataset.** We use the COCO (Caesar et al., 2018) comprising 118K training and 5K validation images for the experiments. For ZPS, we adopt the same setup as PADing (He et al., 2023b) selecting 73 of 80 things and 46 of 53 stuff as seen categories. For ZSS, we follow the setup of DeOP (Han et al., 2023a), choosing 156 of 171 categories as seen categories.

**Implementation Details.** For panoptic segmentation, we leverage MMDetection (Chen et al., 2019), and for semantic tasks, we utilize MMSegmentation (Contributors, 2020) as the code base, and all the experiments are conducted on 8 V100 GPUs. Semantic embeddings are extracted using the CLIP text encoder (Radford et al., 2021) with a ViT-B/16 (Dosovitskiy et al., 2020) backbone. The text templates align with prior works (Ding et al., 2022a; Chen et al., 2023; Zhou et al., 2023; He et al., 2023b). Our base model, Mask2Former (Cheng et al., 2022), employs a ResNet-50 (He et al., 2016) as backbone and trains for 48 epochs with the same setting as the original Mask2Former. The class projector is a simple MLP, and the generator is a VAE structure consisting of four layers of MLP-Batchnorm1d-leakyReLU for the encoder and decoder. In Generation-to-Semantic alignment, the generator trains for 12 epochs. Other hyperparameters are in the ***Supplementary materials***.

**Evaluation Metric.** We follow the GZSL settings (He et al., 2023b; Ding et al., 2022a; Zhou et al., 2023; Chen et al., 2023) where both the seen and the unseen categories need to be segmented correctly. To comprehensively consider the performance for both seen and unseen categories, we apply the harmonic panoptic quality (hPQ) for panoptic segmentation and Intersections over Union (hIoU) for semantic segmentation as the evaluation metric, $hPQ = \frac{2 \cdot sPQ \cdot uPQ}{sPQ + uPQ}$, and $hIoU = \frac{2 \cdot sIoU \cdot uIoU}{sIoU + uIoU}$, where $sPQ$ and $uPQ$ denote the panoptic quality (Kirillov et al., 2019) for the seen and unseen categories. $sIoU$ and $uIoU$ denote the mIoU for the seen and unseen categories. For the inference speed, the metric is Frame Per Second (FPS) with one V100.

### 4.2 Comparison with other methods

**Comparison on Zero-Shot Panoptic Segmentation (ZPS).** We first compare our method with the state-of-the-art ZPS method (He et al., 2023b) in both inductive and transductive settings, presenting the results in Table 2. In the inductive scenario, where we directly inference on the test dataset without finetuning with the synthetic unseen visual features, we observe that PADing achieves 0 hPQ, with its sPQ surpassing ours by 2.8%, attributed to overfitting on the seen categories. However, our approach excels, surpassing PADing by **14%** in uPQ, showcasing the merits of vision-to-cLIP

Table 1: Comparison on Zero-shot Semantic Segmentation. The best results are noted in **bold** and the second best is noted as underline.

| Method | Backbone | Embed | hIoU | sIoU | uIoU | FPS |
|---|---|---|---|---|---|---|
| SPNet (Xian et al., 2019) | | Word2vec | 14.0 | 35.2 | 8.7 | - |
| ZS3 (Bucher et al., 2019) | | Word2vec | 15.0 | 34.7 | 9.5 | - |
| CaGNet (Gu et al., 2020) | ResNet-101 (He et al., 2016) | Word2vec | 18.2 | 33.5 | 12.2 | - |
| SIGN (Cheng et al., 2021b) | | Word2vec | 20.9 | 32.2 | 15.5 | - |
| Zzseg (Xu et al., 2022) | | CLIP | 8.7 | 38.7 | 4.9 | 1.11 |
| ZegFormer (Ding et al., 2022a) | | CLIP | 27.2 | 37.4 | 21.4 | 6.69 |
| ZegCLIP (Zhou et al., 2023) | ViT-B (Dosovitskiy et al., 2020) | CLIP | 40.8 | 40.2 | **41.1** | - |
| PADing (He et al., 2023b) | ResNet-50 (He et al., 2016) | CLIP | 30.7 | 40.4 | 24.8 | - |
| Ours | | CLIP | 36.1 | 40.1 | 32.9 | **24.0** |
| DeOP (Han et al., 2023a) | ResNet-101c (Chen et al., 2018) | CLIP | 38.2 | 38.0 | 38.4 | 4.37 |
| Ours | | CLIP | **41.4** | **42.4** | 40.5 | 9.24 |

Table 2: Comparison on Zero-Shot Panoptic Segmentation.

| Model | Inductive | | | Transductive | | |
|---|---|---|---|---|---|---|
| | hPQ | sPQ | uPQ | hPQ | sPQ | uPQ |
| PADing (He et al., 2023b) | 0.0 | **43.3** | 0.0 | 22.3 | **41.5** | 15.3 |
| Ours | **20.8** | 40.5 | **14.0** | **27.5** | 39.2 | **21.2** |

alignment. In the transductive settings, our method outperforms PADing by 5.2% hPQ and 5.9% uPQ with slightly lower sPQ, showing the merits of generation-to-semantic alignment.

**Comparison on Zero-shot Semantic Segmentation (ZSS).** We conduct experiments on challenging ZSS tasks. The metric we use is hIoU in DeOP (Han et al., 2023a) and the dataset is COCO (Caesar et al., 2018) with only 156 of 171 categories during training. It's important to note that the reported performance does not involve self-training or model ensemble with the CLIP visual encoder. As depicted in Table 1, in comparison to PADing, our sIoU performance is slightly lower (40.1% vs. 40.4%). However, our uIoU significantly surpasses PADing by 8.1%. The hIoU, representing the overall performance of both sIoU and uIoU, is 5.4% higher than PADing. It's worth mentioning that these results are achieved using the ResNet-50 (He et al., 2016), and our method even outperforms some approaches, such as ZegFormer (Ding et al., 2022a) with ResNet-101. When changing to larger backbones, compared with DeOP (Han et al., 2023a), we can achieve higher performance while $2\times$ faster, highlighting our effectiveness. When compared with some methods that are well-designed only for zero-shot segmentation, *i.e.*, ZegCLIP (Zhou et al., 2023), though the mask2former may have weakness in semantic segmentation, we can still achieve higher performance in hIoU (41.4% vs. 40.8%). At the same time, the backbone we use is ResNet101-c (Chen et al., 2018) which is also weaker than ViT-B.

**Comparison on cross-dataset.** To better show the generalization ability of our method, we conduct experiments under the same cross-dataset settings as other methods (Han et al., 2023a; Xu et al., 2022) where only part of the categories are used in training and we directly infer on other datasets without fine-tuning on another dataset. The metric is the mIoU of all the categories, and the results are shown in Table 3. Compared with the SOTA methods, *i.e.*, DeOP (Han et al., 2023a), our method achieve higher performance in PASCAL VOC (VOC-20) (Everingham et al., 2015), and Pascal Context (PC-59) (Mottaghi et al., 2014). Despite slightly lower performance on ADE20k (A-150) due to CLIP integration, DeOP sacrifices speed which is only 50% of our efficiency.

### 4.3 ABLATION STUDY

We employ MaskFormer (Cheng et al., 2021a) as the base model trained 12 epochs in all stages and only on panoptic segmentation due to its difficulty. Unless stated, the ablation studies on our method are conducted under both inductive and transductive settings.

**Ablations on the proposed methods.** The results are shown in Table 4. We set the methods that only use $\mathcal{L}_{match}$ and $\mathcal{L}_{mmd}$ as the baseline. In the inductive setting, though the baseline can generalize to the unseen classes, its performance is too weak. When we adapt the model with synthetic unseen visual features, both sPQ and uPQ drop, indicating that the realistic generation is significant. Then,

Table 3: Comparison on cross-dataset generalization performance.

| Method | Backbone | Training Dataset | VOC-20 | PC-59 | A-150 | FPS |
|---|---|---|---|---|---|---|
| ZS3 (Bucher et al., 2019) | | PASCAL VOC | 38.3 | 19.4 | - | - |
| LSeg (Li et al., 2022a) | | PASCAL VOC | 47.4 | - | - | - |
| OpenSeg (Ghiasi et al., 2022) | ResNet-101 (He et al., 2016) | COCO | 60.0 | 36.9 | 15.3 | - |
| OpenSeg (Ghiasi et al., 2022) | | COCO + Loc. Narr. | 63.8 | 40.1 | 17.5 | - |
| Zzseg (Xu et al., 2022) | | | 88.4 | 47.7 | 20.5 | 1.11 |
| DeOP (Han et al., 2023a) | ResNet-101c (Chen et al., 2018) | COCO-Stuff-156 | 91.7 | 48.8 | **22.9** | 4.37 |
| Ours | | | **93.7** | **49.7** | 20.9 | **9.24** |

Table 4: Ablations on our modules where "V2C" indicates the Vision-to-CLIP alignment, "G2S" indicates the Generation-to-Semantic aligment, and "CPA" indicates the class projector adaptation. Without CPA, the G2S module cannot contribute effectively to performance because the synthetic features for unseen classes cannot be utilized.

| method | Inductive | | | Transductive | | |
|---|---|---|---|---|---|---|
| | hPQ | sPQ | uPQ | hPQ | sPQ | uPQ |
| Baseline | 10.9 | 31.0 | 6.6 | 9.4 | 26.0 | 5.7 |
| Baseline + V2C | **15.8** | **31.2** | **10.5** | 12.3 | 19.7 | 9.0 |
| Baseline + V2C + G2S & CPA | **15.8** | **31.2** | **10.5** | 22.6 | **31.0** | **17.7** |

Table 5: Ablations on target attention and instance alignment in V2C alignment.

| Target Attention | Instance Alignment | Inductive | | | Transductive | | |
|---|---|---|---|---|---|---|---|
| | | hPQ | sPQ | uPQ | hPQ | sPQ | uPQ |
| - | - | 15.4 | **31.2** | 10.2 | 20.3 | 30.1 | 15.3 |
| - | ✓ | 14.0 | 31.0 | 9.0 | 20.9 | 30.5 | 15.8 |
| ✓ | - | 14.5 | 31.0 | 9.4 | 22.0 | **31.0** | 17.0 |
| ✓ | ✓ | **15.8** | **31.2** | **10.5** | **22.6** | **31.0** | **17.7** |

we add the V2C alignment to the baseline and find that though the sPQ does not change too much, the uPQ increases from 6.6% to 10.5%, nearly 4%, indicating the effectiveness of the V2C alignment. If we continue to use this backbone with the baseline generator, we can find that the same performance drop happens. Note that without the class projector adaptation, simply training a generator can not affect the transductive performance as the synthetic unseen features can not be used. Therefore, finally, we add all the rest proposed methods, *i.e.*, G2S and CPA, we find that uPQ achieves a huge improvement from 10.5% to 17.7%.

**Ablations on the components in V2C alignment.** We set the model without target attention and instance alignment as the baseline. Table 5 shows the performance ablation. First, we only add instance alignment to the baseline and find a performance drop in inductive settings due to the drop of unseen classes. However, this loss outperforms the baseline in transductive settings with an improvement in hPQ and uPQ. Next, we replace the instance alignment with the target attention which aggregates all the visual features, and find that though the performance is still lower than the baseline in inductive settings, it is higher than that with only instance alignment. When coming to transductive settings, highlighted feature aggregation receives a large improvement in hPQ and uPQ which are 1.7% higher than the baseline. When combining them, we can find the inductive performance is higher than the baseline and further increases the transductive performance to 2.3%.

**Ablations on the components in G2S alignment.** We also conduct experiments to validate the effectiveness of each component in G2S alignment as shown in Table 6. We set the methods without G2S alignment as the baseline. First, we add $\mathcal{L}_{gv2s}$ to the baseline and find huge performance improvements for all hPq, sPQ, and uPQ which increase 11.3%, 18.5%, and 7.8% respectively. Then we only add the $\mathcal{L}_{g2v}$, though the performance also increases, it is still lower than the $\mathcal{L}_{g2v}$.

Table 6: Ablations on G2V and GV2S in G2S alignment.

| $\mathcal{L}_{g2v}$ | $\mathcal{L}_{gv2s}$ | Transductive | | |
|---|---|---|---|---|
| | | hPQ | sPQ | uPQ |
| - | - | 10.6 | 12.6 | 9.1 |
| - | ✓ | 21.9 | **31.1** | 16.9 |
| ✓ | - | 20.8 | 30.6 | 15.8 |
| ✓ | ✓ | **22.6** | 31.0 | **17.7** |

**Visualization of the generated class features.** To show the quality of the generated visual features, we use t-SNE (Van der Maaten & Hinton, 2008)

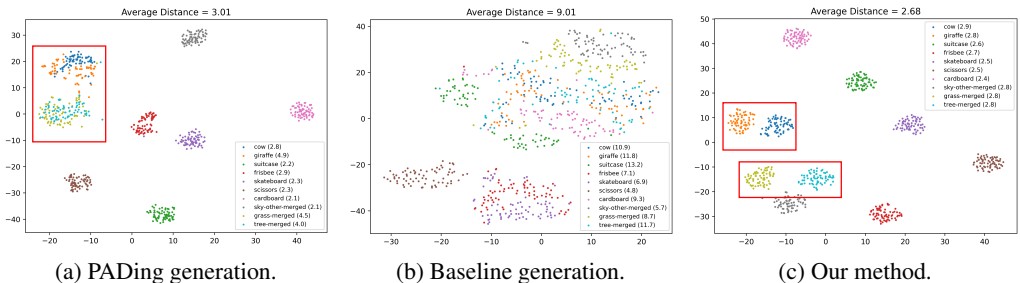

(a) PADing generation.     (b) Baseline generation.     (c) Our method.

Figure 4: T-SNE visualization of the generated visual features for unseen classes: (a) PADing generations, (b) baseline generation, (c) our method's generation. The number in the title represents the overall average distance between all class centers, while the numbers in the legend indicate the distances between individual class centers.

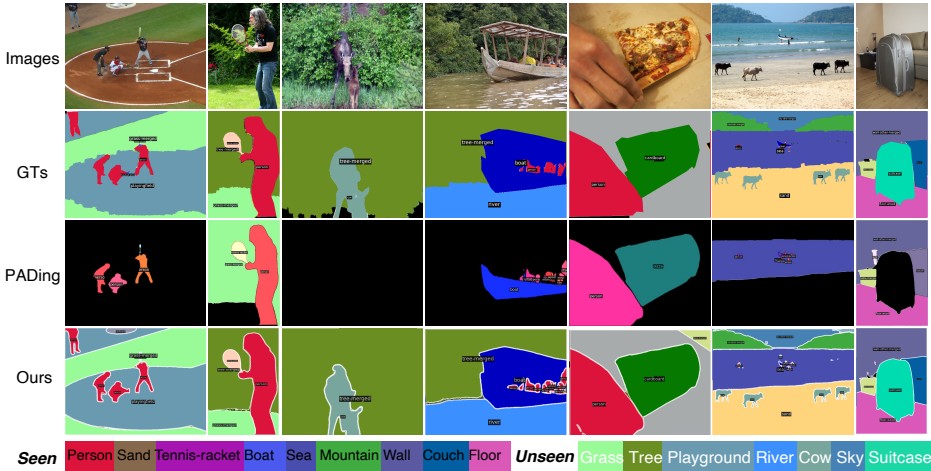

Figure 5: Visualization of our proposed methods. The row column shows the input images and the following rows are the labels, PADing visualizations, and our visualizations.

to visualize their distribution as shown in Fig. 4. Randomly selecting 10 out of 14 unseen categories from ZPS, we first visualize PADing generation, and find that some categories with similar semantics are mixed, *e.g.*, tree and grass. For our baseline method, we can find that the generated visual features are nearly inseparable. Our method achieves the best discriminative abilities and high visual diversity. Due to the space limits, the visualization of the methods without $\mathcal{L}_{div}$ is shown in the **Supplimentary Materials**.

**Prediction visualization.** To qualitatively express the merits of the proposed methods, we visualize the predictions of our proposed methods as shown in Fig. 5. As can be seen in this figure, our method can correctly segment the seen categories, *e.g.*, boat, and the unseen categories which may be missed, *e.g.*, cow, in the third image. More visualization results are in the **Supplimentary Materials**.

## 5 CONCLUSION

In this work, we presented a novel semantic-centric alignment approach for zero-shot segmentation that addresses the limitations of vision-centric methods in handling unseen classes. By aligning generated features with a structured semantic distribution across all classes, we ensured better generalization and reduced overfitting to seen classes. Our two-stage alignment process and the projector adaptation, integrating Vision-to-CLIP alignment and generator training alignment, enriched the semantic features with diverse visual attributes while preserving semantic consistency. This strategy significantly improved the model's capacity to handle unseen classes, achieving state-of-the-art results in both zero-shot panoptic and semantic segmentation tasks. Our approach demonstrates that maintaining a unified semantic-aligned feature space can effectively support zero-shot segmentation without the need for fine-tuning.

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

## A    APPENDIX

You may include other additional sections here.

