# Supplimentart Materials for Semantic-Centric Alignment for Zero-shot Panoptic and Semantic Segmentation

## 1 More Experiment Details

**Task Formulation.** We first define zero-shot segmentation. Given a dataset $\mathcal{D} = \left\{ \mathbf{X}^i, \mathbf{Y}^i \right\}_{i=1}^{M}$ and the semantic embeddings $\mathbf{A} \in \mathbb{R}^{N \times C}$, where $C$ is the channel dimension and $N$ is the number of categories in $\mathcal{D}$. Here, $\mathbf{X} \in \mathbb{R}^{H \times W \times 3}$ represents the images with height $H$ and width $W$. $\mathbf{Y}$ denotes the corresponding pixel-level annotation for $\mathbf{X}$, and $M$ is the number of images in the dataset. The semantic embeddings $\mathbf{A}$ are divided into two subsets: seen categories $\mathbf{A}_s \in \mathbb{R}^{N_s \times C}$ and unseen categories $\mathbf{A}_u \in \mathbb{R}^{N_u \times C}$, where $\mathbf{A}_s \cap \mathbf{A}_u = \varnothing$. Here, $N_s$ and $N_u$ represent the number of seen and unseen categories, respectively. During inference, the model is tested on both seen and unseen classes jointly. In Zero-shot Semantic Segmentation (ZSS) (Ding et al., 2022; Zhou et al., 2023; Chen et al., 2023), images may contain segments belonging to unseen categories $\mathbf{A}_u$. However, in Zero-shot Panoptic Segmentation (ZPS), images only containing segments belonging to seen categories $\mathbf{A}_s$ can be used for training. Additionally, ZPS requires distinguishing each instance in the "thing" categories (Kirillov et al., 2019; He et al., 2023). This different setting leads to an absent data scenario where a small fraction of data is retained and makes ZPS more challenging. Inspired by ZSS (Zhou et al., 2023; Chen et al., 2023), we further divide ZPS into two settings: inductive ZPS, where neither $\mathbf{A}_u$ nor images containing $\mathbf{A}_u$ can be accessed during training, and transductive ZPS, where both $\mathbf{A}_s$ and $\mathbf{A}_u$ are accessible during training, but images containing $\mathbf{A}_u$ remain inaccessible.

**Full experiment settings.** For panoptic segmentation, we leverage MMDetection (Chen et al., 2019), and for semantic tasks, we utilize MMSegmentation (Contributors, 2020) as the code base and all the experiments are conducted on 8 V100 GPUs. Semantic embeddings are extracted using the CLIP text encoder (Radford et al., 2021) with a ViT-B/16 (Dosovitskiy et al., 2020) backbone. The text templates align with prior works (Ding et al., 2022; Chen et al., 2023; Zhou et al., 2023; He et al., 2023). Our base model, Mask2Former (Cheng et al., 2022), employs a ResNet-50 (He et al., 2016) backbone with the same hyperparameters as the original Mask2Former. The class projector is a simple MLP, and the generator is a VAE structure consisting of four layers of MLP-Batchnorm1d-leakyReLU for the encoder and decoder. The generator optimizer aligns with the base model but with a learning rate ten times higher. In union-finetuning, the class projector's learning rate is set at 0.1 times the base model. The base model undergoes default training for 48 epochs (140,350 iterations), less than the original Mask2Former (368,750 iterations). In CAT, the model trains for 12 epochs (36,875 iterations). Hyperparameters include $\gamma = 2$, $\tau = 0.07$, $\lambda = \{2, 5, 10, 20, 40, 60\}$, $\lambda_r = 0.1$, $\lambda_f = 0.01$, bank size in CGA is 32. During inference, the logit for unseen categories is incremented by 1, and no other increments for transductive settings. For union-finetuning, we only report the best performance as different settings lead to different convergence speeds.

## 2 More Experiments

**The number of negative pairs in V2C alignment.** We conduct experiments on the size of the token bank as shown in Table 1. For the inductive settings, we set the size of the bank to 16, 32, and 64. The hPQs achieve 16.2%, 15.8%, and 14.6%, respectively due to the descending uPQs. In the transductive settings, with a bank size of 32, hPQ reaches its peak at 22.6%, accompanied by a slight decrease in sPQ and a substantial improvement in uPQ. When we ablate the bank, the hPQ in inductive drops due to the decrease of uPQ. The hPQ for transductive is good.

Table 1: Ablations on the number of CLS tokens serving as the negative pairs in V2T alignment.

| # Pairs | Inductive | | | Transductive | | |
|---|---|---|---|---|---|---|
| | hPQ | sPQ | uPQ | hPQ | sPQ | uPQ |
| 0 | 15.3 | **31.6** | 10.1 | 22.5 | 30.9 | **17.7** |
| 16 | **16.2** | 31.1 | **10.9** | 19.8 | 30.4 | 14.7 |
| 32 | 15.8 | 31.2 | 10.5 | **22.6** | **31.0** | **17.7** |
| 64 | 14.6 | 30.9 | 9.6 | 20.7 | 30.4 | 15.7 |

Table 2: Ablation studies on $\mathbf{V}^{\varnothing}$.

| $\mathbf{F}^{\varnothing}$ | Inductive | | | Transductive | | |
|---|---|---|---|---|---|---|
| | hPQ | sPQ | uPQ | hPQ | sPQ | uPQ |
| w/o | | | | 22.1 | 31.0 | 17.2 |
| w | 15.8 | 31.2 | 10.5 | **22.6** | 31.0 | **17.7** |

Table 3: Ablation studies on CLS bank size and $\tau_r$.

| Bank Size | $\tau_r$ | Inductive | | | Transductive | | |
|---|---|---|---|---|---|---|---|
| | | hPQ | sPQ | uPQ | hPQ | sPQ | uPQ |
| 16 | 0.07 | **16.2** | 31.1 | **10.9** | 19.8 | 30.4 | 14.7 |
| | 0.15 | | | | 22.0 | 31.1 | 17.1 |
| 32 | 0.07 | | | | **22.6** | 31.0 | **17.7** |
| | 0.15 | 15.8 | **31.2** | 10.5 | 22.0 | **31.1** | 17.0 |
| | 0.30 | | | | 22.1 | 31.0 | 17.2 |
| 64 | 0.07 | | | | 20.7 | 30.4 | 15.7 |
| | 0.15 | 14.6 | 30.9 | 9.6 | 20.9 | 30.7 | 15.9 |

**Ablation studies on the $\mathbf{F}^{\varnothing}$ in embedding contrast.** We perform experiments to validate the efficacy of $\mathbf{V}^{\varnothing}$, as illustrated in Table 2. Our findings reveal that under transductive settings, our approach yields superior hPQ metrics compared to counterparts lacking $\mathbf{F}^{\varnothing}$, owing to the augmented uPQ. This elevation in uPQ stems from the more distributed visual data trained by CAT.

**Ablation studies on the $\tau_r$.** One of the most critical hyperparameters in embedding contrast is $\tau_r$, which regulates the distribution of pseudo-visual embeddings. The results are depicted in the third to fifth line of Table 3. It can be observed that when the $\tau_r$ is set to 0.07, *i.e.*, the visual embeddings are widely distributed, and the hPQ peaks at 22.6%. However, as the $\tau_r$ is increased, leading to a tighter distribution of visual embeddings, the hPQ begins to decline to 22.0% due to a reduction in uPQ. This experiment highlights the benefits of maintaining distributed visual embeddings.

**Experiments on the relationship of $\tau_r$ and CLS token bank.** During experiments, we find an interesting fact that different bank size needs different temperatures to achieve the highest performance as shown in Table. 3. When we set the bank size to 16, we choose two different temperatures, *i.e.*, 0.07 and 0.15, and find a huge performance gap. Precisely, when $\tau_r$ is 0.07 the hPQ is 19.8% and when $\tau_r$ is 0.15 the hPQ is 22.0%, which indicates the gap is 2.2% in hPQ. The gap attributes for both sPQ and uPQ. When the bank size is set as 64, the same phenomenon can be observed, however, the gap is not so big. This experiment shows that the bank size and the temperatures need careful design.

**Experiments on the num of pseudo visual seen embeddings in union-finetuning.** In the union-finetuning stage, we do not generate pseudo **seen** visual embeddings. This experiment validates how much the pseudo visual embeddings will affect the performance as shown in Table 4. As we increase the number of pseudo seen embeddings, the hPQ decreases due to the decrease of uPQ. Meanwhile, the sPQ also decreases slightly. From this experiment, we can conclude that first there is still a large gap between the real and pseudo embeddings. The pseudo seen and unseen visual embeddings are not separated enough.

**Experiments on the num of pseudo visual unseen embeddings in union-finetuning.** We conduct experiments on the sampling number of noise in union finetuning as shown in Table. 5. First, we increase the sampling number from 50 to 300 to produce more pseudo unseen visual embeddings.

Table 4: Ablations on pseudo seen embeddings in union finetuning.

| Num | hPQ | sPQ | uPQ |
|---|---|---|---|
| 0 | **22.6** | **31.0** | **17.7** |
| 10 | 20.7 | 30.8 | 15.6 |
| 50 | 20.0 | 30.9 | 14.7 |
| 150 | 17.7 | 30.7 | 12.4 |

Table 5: Ablation studies on sampling number of pseudo unseen embeddings.

| Num | hPQ | sPQ | uPQ |
|---|---|---|---|
| 50 | 22.3 | **31.0** | 17.5 |
| 150 | **22.6** | **31.0** | 17.7 |
| 300 | **22.6** | **31.0** | 17.8 |
| 600 | 22.1 | 31.0 | 17.1 |

Table 6: Ablation studies on increment in the inductive settings.

| Num | hPQ | sPQ | uPQ |
|---|---|---|---|
| 0.0 | 6.0 | 30.6 | 3.3 |
| 1.0 | 15.8 | **31.2** | 10.5 |
| 1.5 | **17.7** | 30.7 | **12.4** |
| 2.0 | **17.1** | 27.9 | 12.3 |

Table 7: Ablation studies on $\gamma$ in V2C alignment.

| $\gamma$ | Inductive | | | Transductive | | |
|---|---|---|---|---|---|---|
| | hPQ | sPQ | uPQ | hPQ | sPQ | uPQ |
| 1.5 | 13.3 | **31.4** | 8.4 | 20.2 | **31.2** | 14.9 |
| 2.0 | 15.8 | 31.2 | 10.5 | **22.6** | 31.0 | 17.7 |
| 3.0 | **16.0** | 30.5 | **10.8** | 22.4 | 30.1 | **17.9** |

Table 8: Ablation studies on enhancement in V2C alignment.

| Enhancement | Inductive | | | Transductive | | |
|---|---|---|---|---|---|---|
| | hPQ | sPQ | uPQ | hPQ | sPQ | uPQ |
| multiply | **15.8** | 31.2 | **10.5** | **22.6** | 31.0 | **17.7** |
| plus | 14.5 | **31.4** | 9.4 | 22.3 | **31.1** | 17.4 |
| minus | 14.1 | **31.4** | 9.1 | 21.0 | 30.8 | 15.9 |

The hPQ and uPQ increases to their peak of 22.6% and 17.8%. However, when we further increase the sampling number, we observe that the hPQ and uPQ drop to the lowest.

**Experiments on the increments in the inductive settings.** We conduct experiments on how much the increment of the unseen category will affect the performance as shown in Table. 6. As can be seen, we first add nothing to the logits of unseen categories, the hPQ is low due to the low performance of uPQ.

**Experiments on the $\gamma$ in V2C alignment.** We conduct experiments on how much the visual embeddings with segments should be enhanced as shown in Table. 7. As can be seen, in inductive settings, when we increase the enhancement level, though the hPQ increases, the sPQ drops more drastically than the increment of uPQ. In transductive settings, the same phenomenon can be observed and becomes even more apparent. Formally, the hPQ of $\gamma = 2.0$ is higher than $\gamma = 3.0$ due to the large gap in the sPQ. To achieve the balance between the inductive and transductive settings, the $\gamma$ should also be considered.

**Experiments on the enhancement in V2C alignment.** We conduct experiments on how the segments are enhanced as shown in Table. 8. As can be seen, in inductive settings, multiply achieves the highest performance. Plus and minus achieve similar hPQ, however, they can achieve higher sPQ. In the transductive settings, multiply and plus can achieve similar hIoU, and their performance is much higher than minus. This experiment indicates enhancing rather than reducing the embeddings with segments is beneficial for the performance.

## 3 OTHER SETTINGS IN EXPERIMENTS

We apply the same settings as the original settings as MaskFormer Cheng et al. (2021) and Mask2Former Cheng et al. (2022). For all the two tasks and three stages, the images are randomly flipped with a probability of 0.5, and the batch size is set to 16. In ZPS, the image is resized with the scale from 0.1 to 2.0 and cropped to $1024 \times 1024$. In ZSS, the images are cropped to $512 \times 512$ and trained in 80K iterations for all three stages. We use AdamW Loshchilov & Hutter (2017) as the optimizer with an initial learning rate of $10^{-4}$ for ResNet He et al. (2016). Meanwhile, all the ResNet is pre-trained on ImageNet1K Russakovsky et al. (2015).

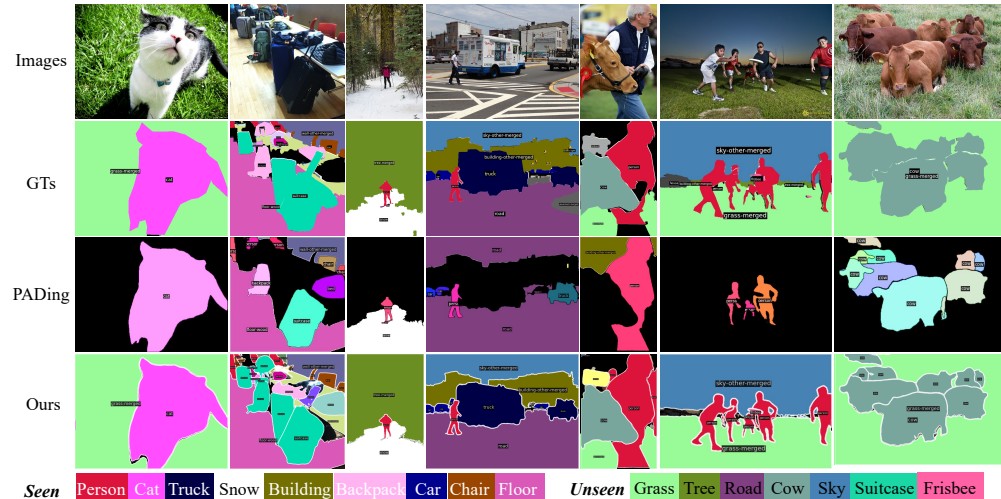

Figure 1: Visualization of our proposed methods in ZPS. The first row shows the input images and the following columns are the labels, PADing's, and ours.

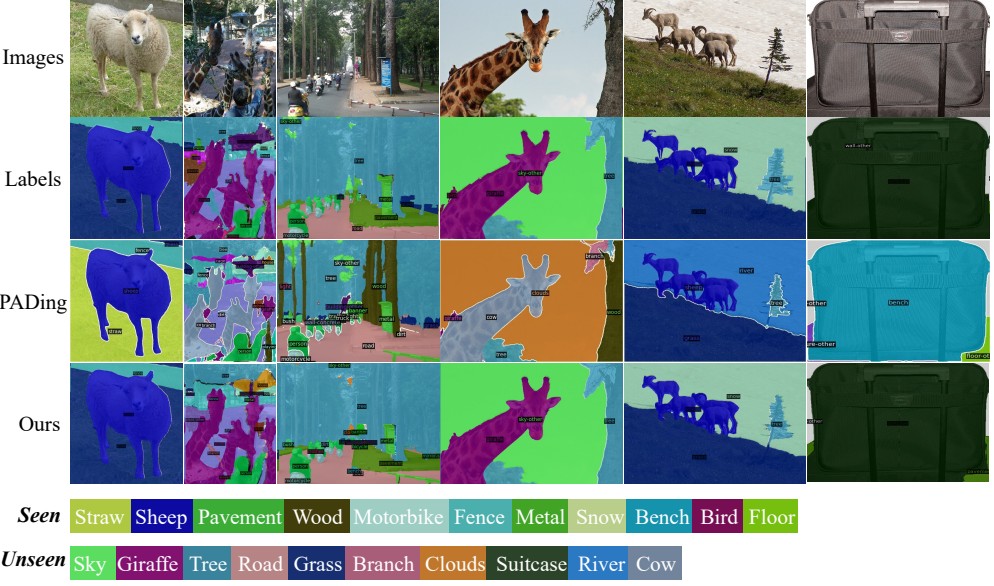

Figure 2: Visualization of our proposed methods in ZSS. The first column shows the input images and the following columns are the labels, PADing's, and ours.

**Unseen categories.** For ZPS, the unseen thing categories are cow, giraffe, suitcase, frisbee, skateboard, carrot, and scissors. The unseen stuff categories are cardboard, sky-other-merged, grass-merged, playing-field, river, road, and tree-merged. For ZSS, the unseen categories are cow, giraffe, suitcase, frisbee, skateboard, carrot, scissors, cardboard, clouds, grass, playing-field, river, road, tree, and wall-concrete.

## 4 MORE VISUALIZATIONS

In this section, we give more visualization results for both ZPS and ZSS. First, we show more results on ZPS as shown in Fig. 1. We can find that our method can segment both seen categories, *e.g.*, cat in the first image and truck in the fourth image, and unseen categories, *e.g.*, sky in the second last image and suitcase in the second image. This figure shows the merits of our methods. In the supplementary materials, we visualize the results for ZSS shown in Fig. 2. We can find that our method can balance the performance for seen and unseen categories.