# OpenReview forum: "Semantic-Centric Alignment for Zero-shot Panoptic and Semantic Segmentation"
_ICLR.cc/2025/Conference — ICLR 2025 Conference Withdrawn Submission_

### Official Review · Reviewer_L3Ks · 2024-10-19

**Soundness:** 2
**Presentation:** 2
**Contribution:** 1
**Rating:** 5
**Confidence:** 4

**Summary:**

This paper addresses the task of zero-shot semantic segmentation, where models trained on base class (seen class) are studied to adapt to novel classes (unseen classes) during testing. These models can easily over-fit to base class and under-perform on novel classes. To mitigate such issue, this paper proposes vision-to-clip alignment, a distillation approach that enables segmentation model like MaskFormer to mimic CLIP representations. To validate effectiveness of their design, they carry out experiments on semantic and panoptic segmentation.

**Strengths:**

[1] Experiments are sufficient. This paper has presented with two recent published papers in zero-shot semantic segmentation, and carry out most needed ablation studies on their designs. Empirically, this paper has validated the effectiveness of distillation approach.

**Weaknesses:**

[1] Technical novelty of “Vision-to-CLIP alignment” is limited. On one hand, aligning dense visual features to pre-trained CLIP vision encoder has been well explored in open-vocabulary field [A] in recent years as a common practice for follow-up papers [B,C,D], as presented in Table 10 a published survey paper [E], where authors are suggested to differentiate the proposed V2C alignment with these approaches. On the other hand, generating synthesized visual representation from word embeddings is also a common practice in zero-shot semantic segmentation [G], where authors are suggested to clarify difference as well.

[2] Unfair comparison to recent work PADing [F]. It should be pointed out that PADing [F] uses CLIP text embeddings, while this paper additionally uses CLIP Vision Encoder for knowledge distillation. A suggested experiment is to include an ablation experiment, where only G2S and CPA are included. Namely, baseline + G2S & CPA. This experiment is missing from ablation table 1. How does this experimental setup performs as compared to PADing [F].

[3] Typos and unclear presentations, as given by Questions section. A more rigorous check on typos is expected.

[A] Open-vocabulary Object Detection via Vision and Language Knowledge Distillation. ICLR 2021.

[B] Learning to Prompt for Open-Vocabulary Object Detection with Vision-Language Model. CVPR 2022.

[C] Detecting Twenty-thousand Classes using Image-level Supervision. ECCV 2022.

[D] F-VLM: Open-Vocabulary Object Detection upon Frozen Vision and Language Models. ICLR 2023.

[E] Vision-Language Models for Vision Tasks: A Survey. TPAMI.

[F] Primitive Generation and Semantic-related Alignment for Universal Zero-Shot Segmentation. CVPR 2023.

[G] Zero-Shot Semantic Segmentation. NeurIPS 2019.

**Questions:**

- In Figure 1 (left) “conventional vision-centric alignment”, what “vision model” refers to is not clear. Meanwhile, which papers are within scope of “conventional vision-centric alignment”? Authors are suggested to use a representative approach instead to replace Figure 1 (left) and avoid confusion. Also, “seen” and “unseen” are vital concepts in zero-shot segmentation, which may differentiate this paper from others. Visualizations of such terms are suggested to include in Figure 1.
- Typo, unclear or inaccurate definition.
    - line 37. suggest a revision from “on seen data” to “on seen classes”.
    - line 41. suggest a revision from “adapting the mododel” to “adapting the model”.

---

### Official Review · Reviewer_iCnp · 2024-11-02

**Soundness:** 3
**Presentation:** 3
**Contribution:** 2
**Rating:** 5
**Confidence:** 3

**Summary:**

This paper presents a novel approach for zero-shot segmentation by introducing a semantic-centric alignment method aimed at improving generalization to unseen classes. The proposed approach aligns features with a well-structured semantic distribution across both seen and unseen classes. The method involves Vision-to-CLIP alignment, generating synthetic features for unseen classes, and finetuning the class projector with semantic-aligned joint features. The paper demonstrates state-of-the-art performance on both zero-shot panoptic and semantic segmentation tasks.

**Strengths:**

1. The paper is well-written and easy to understand.
2. The visualizations are extensive, particularly the T-SNE plots, which showcase the method's strong discriminative capabilities and high visual diversity.

**Weaknesses:**

1. My primary concern is with the paper's originality. While semantic alignment with CLIP and generating synthetic features are widely utilized in zero-shot tasks, it would be helpful to clarify what unique contributions this method offers specifically for zero-shot segmentation.
2. The use of a class projector in zero-shot tasks is also a common practice, which does not constitute a major innovation of the paper.
3. In Table 1, ZegCLIP outperforms the proposed method in terms of uIoU, which is particularly relevant to zero-shot tasks that focus on unseen classes. Does this imply that the proposed method may be less effective than ZegCLIP for truly unseen classes?
4. Regarding the comparison methods presented in Table 2, PADing does not seem to support an inductive setting. Should other methods that are compatible with inductive settings be considered instead of a direct comparison to zero?
5. Segment Anything v2 is currently popular in zero-shot segmentation. How does the proposed method compare to it?
6. Although the paper includes comparisons to many methods, it lacks evaluation against the latest SOTA methods from 2024, which would provide a more current performance benchmark.

**Questions:**

Please refer to Weaknesses.

---

### Official Review · Reviewer_Ka1u · 2024-11-04

**Soundness:** 3
**Presentation:** 3
**Contribution:** 2
**Rating:** 6
**Confidence:** 4

**Summary:**

This paper presents a zero-shot segmentation method, which tries to using a semantic-centric strategy instead of vision-centric methods. For better generalization and reducing overfitting to seen classes, the suggested method aligns generated features with a structured semantic distribution across all classes. Two-stage alignment process and the projector adaptation are introduced used to enrich the semantic features with diverse visual attributes while preserving semantic consistency. This paper achieves SOTA performance in both zero-shot panoptic and semantic segmentation.

**Strengths:**

The proposed semantic-centric method is novel as it aligns the backbone and generator features with the well-structured semantic distribution to ensure all classes are properly positioned.

This paper achieves SOTA performance in both zero-shot panoptic and semantic segmentation.

This paper  is well- presentation and provides sufficient detail, making it easy to follow.

**Weaknesses:**

Some descriptions are exaggerated, such as the statement on line 88 about “addressing the overfitting problem.” The experiments show that compared to DeOP, the improvement for unseen classes is lower than that for seen classes, and for other datasets, there seems to be no significant improvement for unseen classes. How can the proposed method be validated to address the problem of zero-shot segmentation methods being prone to overfitting to seen classes?

In the “Visualization of the generated class features” section, I believe that visualizing both the seen class features and the generated unseen class features is more effective in demonstrating discriminativeness than visualizing the carefully selecting 10 unseen classes. This is because both types of classes exist during the inference stage. If the differences between different seen classes are significantly greater than those between unseen classes, it would also lead to difficulties in discrimination.

The authors have conducted ablation experiments on multiple components but have not effectively demonstrated the effectiveness of the semantic-centric strategy. What effect can existing method (such as DeOP) achieve by introducing alignments with a structured semantic distribution across all classes?

**Questions:**

Please refer to weakness

---

### Official Review · Reviewer_MHYL · 2024-11-04

**Soundness:** 3
**Presentation:** 4
**Contribution:** 2
**Rating:** 3
**Confidence:** 5

**Summary:**

The paper presents a novel Semantic-Centric Alignment approach for zero-shot segmentation, specifically targeting zero-shot panoptic and semantic segmentation. Traditional methods often employ a vision-centric alignment approach, which aligns generated visual features to the seen visual distribution, leading to overfitting and poor generalization for unseen classes. To address this limitation, the authors propose a semantic-centric alignment method that leverages the well-structured semantic space of CLIP (Contrastive Language-Image Pretraining). This approach enhances generalization by aligning both the visual backbone and generated features with CLIP’s semantic embeddings, thereby creating a unified and semantically consistent feature space.

**Strengths:**

* The proposed three-stage alignment process is clearly defined and theoretically grounded, enhancing reproducibility.

* This paper is well written and easy to follow.

**Weaknesses:**

* Since zero-shot and open-vocabulary are very similar tasks, and this work also uses CLIP, it would be better to discuss and compare with recent open-vocabulary works.

* Semantic-centric alignment is quite common in open-vocabulary works. The novelty is not enough.

* The proposed methods are very old. Most of them are before 2023. As an ICLR 2025 paper, more recent works in 2024 should be compared, including open-vocabulary works, since zero-shot and open-vocabulary works use similar experimental settings.

* Only CLIP is used in the proposed model. It would be great to use more diverse pre-trained models to demonstrate the generalizability.

**Questions:**

Please see weaknesses.

---

### Note · Authors · 2024-11-13

I have read and agree with the venue's withdrawal policy on behalf of myself and my co-authors.